# A Review of Gallic Acid-Mediated Fenton Processes for Degrading Emerging Pollutants and Dyes

**DOI:** 10.3390/molecules28031166

**Published:** 2023-01-24

**Authors:** Juan Pablo Pereira Lima, Carlos Henrique Borges Tabelini, André Aguiar

**Affiliations:** Institute of Natural Resources, Federal University of Itajubá, Itajubá 37500-903, MG, Brazil

**Keywords:** gallic acid, Fenton reaction, contaminants of emerging concern, brominated flame retardant, pharmaceutical, pesticide

## Abstract

Diverse reducing mediators have often been used to increase the degradation of emerging pollutants (EPs) and dyes through the Fenton reaction (Fe^2+^ + H_2_O_2_ → Fe^3+^ + HO^●^ + HO^−^). Adding reductants can minimize the accumulation of Fe^3+^ in a solution, leading to accelerated Fe^2+^ regeneration and the enhanced generation of reactive oxygen species, such as the HO^●^ radical. The present study consisted in reviewing the effects of gallic acid (GA), a plant-extracted reductant, on the Fenton-based oxidation of several EPs and dyes. It was verified that the pro-oxidant effect of GA was not only reported for soluble iron salts as a catalyst (homogeneous Fenton), but also iron-containing solid materials (heterogeneous Fenton). The most common molar proportion verified in the studies was catalyst:oxidant:GA equal to 1:10–20:1. This shows that the required amount of both catalyst and GA is quite low in comparison with the oxidant, which is generally H_2_O_2_. Interestingly, GA has proven to be an effective mediator at pH values well above the ideal range of 2.5–3.0 for Fenton processes. This allows treatments to be carried out at the natural pH of the wastewater. The use of plant extracts or wood barks containing GA and other reductants is suggested to make GA-mediated Fenton processes easier to apply for treating real wastewater.

## 1. Introduction

Emerging pollutants (EPs) are micropollutants, synthetic or natural, recently considered hazardous to the environment and consequently to the health of living organisms. They are commonly found in soil and aquatic bodies. Some examples of EPs are pharmaceuticals, brominated flame retardants, pesticides, and some components of personal-care products [1]. Although they are not classified as EPs, synthetic dyes are pollutants present in wastewaters from several industries, especially from the textile sector, and they cause several negative impacts on nature [2]. Owing to the ineffectiveness of conventional treatments in regard to degrading such pollutants, advanced oxidation processes (AOPs) have been relevant to their treatment and pretreatment. AOPs are defined as processes that are capable of generating hydroxyl radicals (HO^●^) in sufficient quantities for degrading organic pollutants into biodegradable molecules or completely mineralizing them into CO_2_, H_2_O, and salts. Different types of AOPs, including photolysis, photocatalysis, ozonation, Fenton reaction, and sonochemical and electrochemical oxidation systems, have been recently reviewed for degrading EPs [3] and dyes [4].

In studies comparing different AOPs, those based on the Fenton reaction (Reaction (1)) have stood out due to the higher economic feasibility [5,6] and higher pollutants removal [7,8], as well as the increase of target pollutants biodegradability [8,9,10]. In the Fenton reaction, the ferrous ions catalytically degrade the hydrogen peroxide to generate hydroxyl radical. This radical has a high standard reduction potential (E° = 2.80 V/SHE) and a short lifetime, and it is able to rapidly and indistinctly degrade organic molecules in solution. When using ferric ions, the Fenton-like reaction occurs (Reaction (2)). However, the radical produced is the hydroperoxyl, HO_2_^●^, which has a lower standard reduction potential (E° = 1.44 V/SHE) and consequently lower effectiveness in degrading pollutants in comparison with HO^●^. Despite being very slow, the second reaction is important to reduce Fe^3+^ to Fe^2+^, which enables it to take part in the first reaction again [11]. These reactions require pH control since the ideal range of pH value is considered to be between 2.5 and 3.0 [12,13] because, if it is above 4.0, the iron precipitates as ferric hydroxide, making the catalyst unavailable [14]. After the degradation of an organic pollutant in solution, its neutralization is required to meet the disposal standards and also to separate the iron from the treated wastewater, which can be discharged into receiving water bodies or reused [15,16,17].
Fe^2+^ + H_2_O_2_ → Fe^3+^ + HO^●^ + HO^−^ k = 50–80 mol^−1^.L^−1^(1)
Fe^3+^ + H_2_O_2_ → Fe^2+^ + HO_2_^●^ + H^+^ k = 0.002–0.01 mol^−1^.L^−1^(2)

Another important aspect is the iron and H_2_O_2_ concentration because, if one or both inputs exceed the ideal concentration, unwanted scavenging of HO^●^ radicals (Reactions 3 and 4, respectively) might occur, and this impairs the pollutants’ degradation. Several studies have evaluated the optimum reactants concentration, as well as the pH effect, temperature, type, and physical state of the catalyst, besides interfering substances [7,18,19,20,21]. Regardless of the ideal reaction conditions, the accumulation of Fe^3+^ ions in the solution is common, which must be regenerated to Fe^2+^ to enable a continuous and long-lasting formation of HO^●^ radicals.
Fe^2+^ + HO^●^ → Fe^3+^ + HO^−^ k = 2.5–5 × 10^8^ mol^−1^.L.s^−1^(3)
H_2_O_2_ + HO^●^ → H_2_O + HO_2_^●^ k = 1.7–4.5 × 10^7^ mol^−1^.L.s^−1^(4)

In order to increase the efficiency in the organic pollutants’ degradation through the Fenton reaction and minimize its limitations, different approaches can be integrated into the treatment. The most well-known approaches consist of using UV/visible irradiation (called photo-Fenton), electrical current (electro-Fenton), ultrasound (sono-Fenton), or a combination of them (photo-electro-Fenton, sono-photo-Fenton, or sono-electro-Fenton), which have recently been reviewed for degrading dyes and textile wastewaters [11]. The photo-Fenton process uses UV/visible irradiation (λ < 600 nm) to promote the regeneration of Fe^3+^ to Fe^2+^, which increases the catalytic capacity of the process and decreases the formation of ferrous sludge due to the lower amount of iron required. Meanwhile, H_2_O_2_ photolysis also occurs, enabling another way to produce HO^•^ radicals [22]. The electro-Fenton provides the reaction medium with an electrical current through electrodes. Depending on the type of arrangement of the electrolytic cell and the composition of the electrodes, the cathode can promote *in situ* generation of H_2_O_2_ from O_2_, along with Fe^2+^ regeneration [23]. In sono-Fenton processes, ultrasound promotes the propagation of expansion and compression cycles, generating the acoustic cavitation phenomenon in solution. It tends to promote high temperatures and internal pressures on the microbubble liquid interface that consequently cause the water and H_2_O_2_ sonolysis to produce HO^●^ radical. In addition, ultrasonic irradiation enhances Fe^2+^ regeneration from the complex between the Fe^3+^ ions and hydrogen peroxide (Fe-O_2_H^2+^). Fe^2+^ can also be regenerated by the reaction between Fe^3+^ and H^•^, which is generated through the water sonolysis [24]. Table 1 compiles some advantages and disadvantages of Fenton processes coupled with UV/visible irradiation, electrical current, and/or ultrasound, as highlighted in recently published reviews.

The aforementioned processes similarly require an extra expenditure of energy, making their real application difficult. Another approach to overcome the problems related to Fenton processes consists of the addition of Fe^3+^-reducing phenolic compounds to the reaction medium [25,26]. Certain phenols can constantly promote the regeneration of Fe^2+^ much faster than H_2_O_2_ (Reaction (2)), and they tend to minimize unwanted accumulation of Fe^3+^ generated by the classical Fenton reaction and consequently enable the generation of more HO^●^ radicals without energy consumption [18,27,28,29,30,31]. Several synthetic compounds have been tested and compared as reductants in Fenton processes [25,32,33,34], but many of them present certain toxicity, as is the case of hydroxylamine [35].

Compounds extracted from plants can be a promising alternative in Fenton processes, as they are generally non-toxic and there are no synthesis-related costs to be obtained. Natural compounds can contribute to efficient and sustainable wastewater treatment, with lower energy consumption and lower costs [36,37]. Lignin degradation, an aromatic macromolecule present in plant tissue, generates phenols with reducing properties, which have shown promising results in the decolorization of dyes in Fenton processes [25,38]. In addition to lignin, plants also contain tannins, which are low-molecular-mass polyphenols [39]. Gallic acid (GA) is a tannin precursor molecule with Fe^3+^-reducing activity [25,40], which can be more simply obtained [41] and have a lower cost compared to synthetic reductants. GA has different applications [41], as shown in Figure 1, including as a potential pro-oxidant in Fenton processes.

Figure 2 shows the effects of GA and its intermediates generated by reducing Fe^3+^ ions. As verified by Aguiar and Ferraz [25], the reduction of Fe^3+^ ions to gallic acid is superior to the stoichiometry of 2:1. This demonstrates that GA-derived quinone can also regenerate Fe^2+^ ions, as well as other reducing phenols observed in the same study. Quinone can also be regenerated to its precursor, the semiquinone radical, or mineralized into CO_2_ and H_2_O [18,42]. This last way enables the degradation and mineralization of GA and its oxidized intermediate during the treatment of a possible target pollutant, which minimizes secondary pollution problems, in addition to promoting the desired Fe^2+^ regeneration.

Given its pro-oxidant properties when regenerating Fe^2+^ ions to accelerate/increase HO^●^ radical generation, the present study aimed to review the effects of the reducing phenol originating from plants, gallic acid, as a mediator in the oxidative degradation of EPs and dyes by Fenton processes. Studies addressing different aspects by using this mediator were found, and a broad analysis of the published results is presented here.

## 2. Effect of GA on the Fenton-Based Degradation of EPs and Dyes

Table 2 presents the data compiled from more than 20 studies that evaluated the addition of GA in order to act as a pro-oxidant in the degradation of different EPs and dyes. It is important to emphasize that the tabulated data consist of the main results with and without the addition of GA. Most of the studies evaluated reactions in homogeneous systems, using Fe^2+^ or Fe^3+^ salts as catalysts. Some studies have evaluated copper and vanadium salts as catalysts in the conversion of H_2_O_2_ to free radicals, similar to iron.

The majority of studies evaluated H_2_O_2_ as an oxidant, while less than 20% of them evaluated persulfate (PS) or peroxymonosulfate (PMS). Along with HO^●^, the sulfate radical (SO_4_^2−●^) can be generated in solution from PS and PMS, and it has a higher standard reduction potential than HO^●^ [4,14]. Through these other peroxides, GA has also increased the generation of free radicals and consequently the degradation of target pollutants [20,28,29]. 

### 2.1. Effect of pH

The pH is an important operational parameter in Fenton processes. By adding phenolic reductants, some of them have chelating properties, keeping the Fe ions in solution at pH values higher than the optimal range, which is between 2.5 and 3.0. This aspect dismisses the decrease of pH to the ideal range, and an eventual neutralization after treatment [4]. Dong et al. [28] analyzed the influence of the initial pH on the degradation of iopamidol. The Fe^3+^/PS/GA reaction system removed 80% of the target pollutant at pH values of 4 and 5. By increasing the pH, a decrease in the ability of complexation and reduction of Fe^3+^ by GA was suggested. In addition, there was the precipitation of ferric ions in the form of their corresponding oxy-hydroxides, which decreased the removal of iopamidol to 52%, at pH 9, according to the authors. Huang and Yang [30] evaluated the effect of pH on the degradation of the sulfamethoxazole antibiotic by heterogeneous Fenton/GA. The removal of the target pollutant at pH 4 was 38.1%, and it was practically nullified at pH 6 and 8. According to the authors, it was due to the deprotonation of the catalyst surface at high pH values, impairing the adsorption of GA to interact with Fe ions. Another justification mentioned by them was the instability of the H_2_O_2_ in more alkaline mediums. It is important to outline that heterogeneous Fenton processes are most effective when adsorption of the target pollutant occurs on the catalyst surface for its further degradation [56,57]. 

The influence of the initial pH was also studied by Pan et al. [43] when analyzing the addition of GA in the degradation of an organobromine by Fe^3+^/PS. The reaction conditions in alkaline medium were unfavorable, as less than 50% of the target pollutant was degraded at pH 9 after 72 h of reaction, while degradation of approximately 80% was verified in pH 3, 5, and 7. When studying the degradation of a pesticide mixture by GA-mediated photo-Fenton system, Papoutsakis et al. [44] found an increase of approximately 15% in the mineralization of pesticides by reducing the pH from 5 to 3 in the treatment. On the other hand, when studying the degradation of bisphenol A by Fe^3+^/PMS, Zhang et al. [20] observed an increase in the degradation and reaction rate constants as a function of the increase in pH up to 9.

### 2.2. Effect of Temperature

Regarding different reaction temperatures, Tabelini et al. [19] studied the addition of GA in Fenton processes to oxidize azo dyes. They found that the pro-oxidant effect of the reductant was more expressive at lower temperatures, 20 °C, and 30 °C. When increasing to 40 °C and 50 °C the pro-oxidant effect of GA was faded by the more significant effect of temperature. The authors suggested that the number of collisions between Fe ions and H_2_O_2_ must be stimulated with the increase in temperature, e.g., favoring Reaction (2), and this minimizes the effect of GA in higher temperatures.

In studies evaluating the effect of temperature, it was possible to calculate the activation energy of the different reaction systems. For Methyl orange decolorization, the addition of GA decreased the activation energy by 52% and 29% in reactions initially containing Fe^2+^ and Fe^3+^ as catalysts, respectively. For the azo dye Chromotrope 2 R, which was more susceptible to decolorization, the reductions in activation energy were lower, 2.4% and 12.5%, respectively [19]. In another study by the same group, Lima et al. [52] also reported a decrease in activation energy in Fenton/GA systems when oxidizing Bismarck brown Y dye. By adding GA, the activation energy was reduced by 39% for Fe^2+^/H_2_O_2_ and 49% for Fe^3+^/H_2_O_2_.

### 2.3. Effect of Catalyst Form

Considering the oxidation state of Fe ions, the pro-oxidant effect of GA in Fenton processes was more noticeable in mediums initially containing Fe^3+^ salts compared to Fe^2+^ [19,28,32,42,50,52], because the Fe^3+^ needs to be reduced to start the production of HO^●^ radicals. In addition to soluble salts or solid materials containing iron, for copper and vanadium salts, the use of GA in the reactions also increased the degradation of target pollutants. GA also reduces Cu^2+^ and V^5+^ to Cu^1+^ and V^3+^, respectively, with the latter two being more effective in the conversion of H_2_O_2_ to HO^●^ radical [25,29,49].

By studying the heterogeneous catalyst goethite in the degradation of a pesticide, Lin et al. [58] evaluated the effect of several chelators, including GA. With 5 mmol.L^−1^ of GA, a reduction in the degradation from 56.7% to 21.6% was observed. According to the authors, the adsorption of GA on the catalyst surface might have inactivated the catalytic sites, decreasing the decomposition of H_2_O_2_. Compared to other studies evaluating heterogeneous catalysts and reviewed here, the work of Lin et al. [58] was the only one that reported the negative effect of GA adsorption on the catalyst.

### 2.4. Effect of GA Dosage

Although the reviewed studies highlight the pro-oxidant effects of GA, it is important to evaluate its optimal concentration in different reaction systems. Sousa and Aguiar [32] studied the influence of reductant concentration on the designated reactions in decolorizing a dye. They found that the relationship between reductant concentration and increased decolorization was not proportional since higher decolorization was obtained with lower concentrations of GA, i.e., 10 and 30 µmol.L^−1^, regardless of the initial oxidation state of iron. For higher concentrations of GA, inhibition of decolorization was observed. Similarly, Dong et al. [42] observed that higher concentrations of GA decreased the Fe^3+^/H_2_O_2_ decolorization of Methyl orange. When evaluating the Fe^3+^/H_2_O_2_/pyrophosphate degradation of 2,2′,5-trichlorodiphenyl, Zhao et al. [21] verified an increase in the oxidation of the target pollutant when reducing the GA concentration of 1 mmol.L^−1^ to 0.01 mmol. L^−1^. The three aforementioned studies, in addition to other published ones [18,20,28,43,49], suggest that the excessive reductant competes with the target pollutant for free radicals. This behavior has been commonly observed for other natural reductants, such as 3-hydroxyanthranilic acid [59], vanillin [38], cysteine [60,61], and ascorbic acid [62]. According to Strlič et al. [40], if the molar ratio is less than 2:1 between GA and Fe^3+^ ions, the reductant presents pro-oxidant action, increasing the production of HO^●^; if the molar ratio of GA:Fe is higher, the excessive mediator acts as an antioxidant due to its HO^●^ scavenging activity. In a way, this is a positive aspect: since the reductant is effective in low concentrations, it does not increase the cost of a possible treatment due to its addition in higher concentrations.

Aside from the inhibitory behavior at high concentrations of GA, it was difficult to find a tendency among the different published works due to the different reaction systems tested. Some studies have verified the pro-oxidant effect of GA in a very low concentration of this reductant, 10 µmol.L^−1^, regardless of the concentration of the target pollutant. The most commonly evaluated molar ratio in the studies was catalyst:oxidant:GA equal to 1:10–20:1. It indicates the necessity of low concentrations of both catalyst and gallic acid in the oxidation reactions compared to the required concentration of oxidant, in addition to corroborating that the GA:Fe ratio should be less than 2:1, according to Strlič et al. [40]. The concentration of the oxidant must be higher in order to favor the chemical equilibrium and generate more HO^●^ radicals, but not so high as to scavenge them (Reaction 4).

## 3. Comparison with Other Reductants

Several studies have evaluated different reducing and non-reducing compounds, highlighting GA as one of the most effective mediators in the degradation of EPs and dyes by Fenton processes [25,28,46]. Sun and Pignatello [45] highlighted the chelating ability of GA and the high oxidative activity of the complex Fe-GA. Of the 50 compounds tested by them to solubilize Fe at pH 6 and catalyze the oxidation of a pesticide, Fe^3+^/H_2_O_2_/GA system was the most effective, since it degraded completely the target pollutant at only 10 min. This result was similar to the Fe^3+^/H_2_O_2_ system at pH 2.8. When studying the degradation of an organochlorine, Ma et al. [33] verified that the addition of phenolic reductants at 1 and 10 mmol.L^−1^ had a slight improvement after an 8 h reaction. However, the addition of all reductants at 100 mmol.L^−1^ had an inhibitory effect, except for GA. 

The studies of a same research group have compared the effect of several aromatic reducing compounds in Fenton processes for decolorization of dyes [32,48,50,51,52]. In one of these more recent works, it was verified that synthetic dihydroxybenzenes, such as catechol and hydroquinone, were similarly effective in comparison to GA in removing dyes and increasing the values of the reaction rate constants [52]. In addition to phenolic hydroxyl groups, which are responsible for the Fe^3+^-reducing activity in these three compounds, it is known that other functional groups interfere positively or negatively with their pro-oxidant properties [25,27]. Analogously to the 2,3-dihydroxibenzoic acid [63], the carboxyl group present only in the GA is not oxidized by Fe^3+^, but it facilitates the ion chelation and, consequently, its reduction. However, as the three reducers were similar in the study by Lima et al. [52], it is suggested to apply the GA, as it has the advantage of being a natural compound. 

## 4. Other Alternatives in Using Plant-Derived Reductants in Fenton Processes

Substances related to GA, containing three vicinal hydroxyl groups in the aromatic ring and of plant origin, have also been evaluated as pro-oxidants in Fenton processes. Bu et al. [64] analyzed the effect of epigallocatechin-3-gallate on the Fe^2+^/PS oxidation of atrazine. The mediator showed both chelating and reducing properties by promoting an increase from 29% to 96% in pesticide oxidation. Bolobajev et al. [65] found that the addition of tannic acid, a reductant with a structure much more complex than the GA one, increased the generation of HO^●^ radicals and enabled the complete oxidation of 2,4,6-trichlorophenol by soluble Fe^3+^/H_2_O_2_. In addition, tannic acid promoted the solubilization and reduction of Fe^3+^ from ferric sludge at pH 3, enabling organochlorine to be degraded. Through these results, the authors suggested that residual ferric sludge could be reused as a catalyst in Fenton processes. 

To consider the feasibility of Fenton processes in degrading EPs and dyes present in wastewater, studies on cost analysis should be performed, as recently reviewed [11]. It is worth noting that the addition of GA in Fenton processes is not trivial, since it would involve the cost related to its acquisition, in addition to other inputs. An alternative to use reductants would be to test solutions naturally containing GA and/or other phenols. Treatments with reductants present in plant-derived fluids have been published. One of these first studies evaluated the addition of aqueous *Pinus taeda* wood extracts [66]. For 0.1% *v*/*v* of extracts, the authors detected increments in the oxidation of Azure B dye by Fe^3+^/H_2_O_2_ and Cu^2+^/H_2_O_2_ reaction systems. Through the optimized concentration of 0.03 g.L^−1^ of green tea, used as a source of reductants, Pan et al. [43] verified an increase from 8% to 76% in the Fe^3+^/PMS degradation of an organobromine. 

When using a real wastewater from cork processing as a source of reductants (diluted 2% *v*/*v*), Papoutsakis et al. [44] verified an increase of up to ~70% in the degradation of imidacloprid at pH 3 and 5 by Fe^3+^/H_2_O_2_ and Fe^3+^/H_2_O_2_/UV. The results were attributed to the phenols present in the wastewater, which were able to maintain iron ions in solution, and additionally regenerate Fe^2+^. Manrique-Losada et al. [54] proposed the use of extracts from different fruits, which contain polyphenols, including GA, in the Fe^3+^/H_2_O_2_/UV process. In the presence of cupuaçu extract (containing 2.45 g.L^−1^ of polyphenols), there was a much higher degradation of the four pharmaceuticals tested as targets, above 95%. In addition, the extract of this fruit was responsible for increasing the solubility of Fe in solution and the decomposition of H_2_O_2_ residual, dismissing the removal of this input after treatment. When replacing the deionized water used in the preparation of pollutant solutions by municipal sewage, the reaction system was less effective in the degradation of pharmaceuticals. It suggests that humic substances present in the sewage negatively affected the treatment [54]. It is important to highlight that the two studies aforementioned, as well as others present in Table 2, showed that GA and plant extracts presented pro-oxidant properties also in photo-Fenton processes. It shows that the integration of reductants and UV/visible irradiation increases the degradation of pollutants.

Alternatively, Romero et al. [31] extracted insoluble tannins from the *Pinus radiata* bark as a natural source of reductants that could be recovered after the Fe^3+^/H_2_O_2_ degradation of atrazine. An increase in the generation of hydroxyl radicals by tannins was verified, and the degradation of the target pollutant was 93% at pH 3.4 after 30 min of reaction. Continuing this study, the same group evaluated the stability of this material in the degradation of the same target pollutant [67]. Interestingly, they found no loss of pro-oxidant activity after five cycles of reuse. Moreover, the leaching of polyphenols or other aromatic compounds into the solution was not verified, which would avoid secondary pollution problems. Based on the fundamentals of green chemistry and biorefinery, there has been a growing increase in the use of agricultural waste as a more accessible source of reducing polyphenols [68]. Expressive amounts of polyphenols, including GA, have been found in peanut, rice, coffee, sugarcane, corn, and wheat residues [69], which could be tested as a potential source of reductants in Fenton processes.

Another alternative approach aiming the reuse consists of immobilizing reductants on a support. In this way, Pagano et al. [70] verified an increase of approximately 60% in the Fe/H_2_O_2_ degradation of a nonionic surfactant when using hydroquinone adsorbed on granular activated carbon. Interestingly, only 3% of the reductant was desorbed from the support after treatment. Another way to reuse the reducer supported on a solid material was evaluated by Zhang et al. [57]. They synthesized a stable material containing cysteine as a reductant intercalated in a matrix of layered double hydroxides containing copper and aluminum. In the presence of H_2_O_2_, the prepared catalyst degraded approximately 95% of Rhodamine B and 80% of p-nitrophenol in 60 min of reaction.

## 5. Conclusions and Future Perspectives

This work reviewed several studies that investigated the effects of gallic acid on the oxidation of emerging pollutants (pesticides, brominated flame retardants, and pharmaceuticals) and dyes by Fenton processes. It was observed that, in general, the addition of GA in such reaction systems improvises the degradation of different pollutants. This is justified by the fact that the GA reduces Fe^3+^, constantly regenerating Fe^2+^ for the Fenton reaction, and enabling it to generate more HO^•^ radicals. The most common molar ratio for catalyst: oxidant: GA evaluated in the consulted studies was 1:10–20:1, which indicates the need for low concentrations of catalyst and gallic acid in oxidation reactions compared to the required concentration of oxidant. All studies that evaluated the influence of GA concentration showed better results with low concentrations of the reductant, while for higher concentrations, the reductant was inhibitory. In addition, GA has proven to be an effective mediator at pH values above the ideal range (2.5–3.0) for Fenton processes, enabling wastewater treatment to be carried out at its natural pH without any pH correction. Promising results with plant extracts or wood barks containing GA, as well as other reductants, have also been published. On the other hand, an important aspect that needs evaluation is the toxicity resulting from the treated samples due to the use of GA in Fenton processes. Most of the studies reviewed here focused on the degradation of a target pollutant in synthetic wastewater. It shows that studies are needed to treat real wastewater, especially those containing EPs and dyes, which are mixed with other pollutants. By treating real wastewaters, it is possible to assess whether GA-mediated Fenton processes meet discharge standards in receiving water bodies and/or reuse standards. If reusing treated wastewaters in the industrial plant, water intake can be minimized, thus allowing for the conservation of natural sources and reducing the costs of the process. It is also important to highlight that all reviewed works performed only bench-scale treatments and mainly in batch systems. Pilot-scale experiments should be encouraged, in addition to technical and economic feasibility analysis. To sum up, further studies are needed to improve Fenton processes mediated by gallic acid.

## Figures and Tables

**Figure 1 molecules-28-01166-f001:**
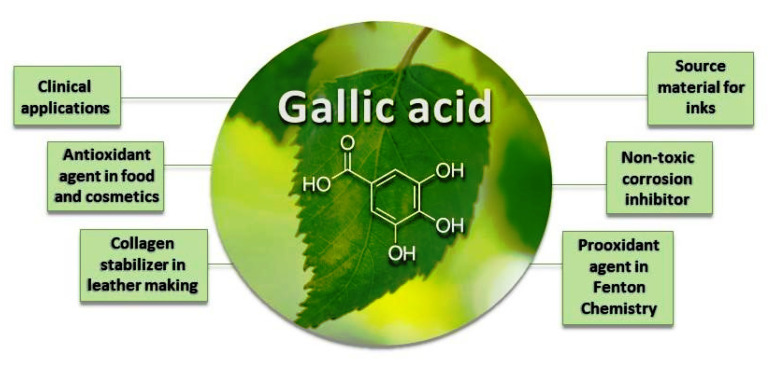
Different applications for gallic acid.

**Figure 2 molecules-28-01166-f002:**
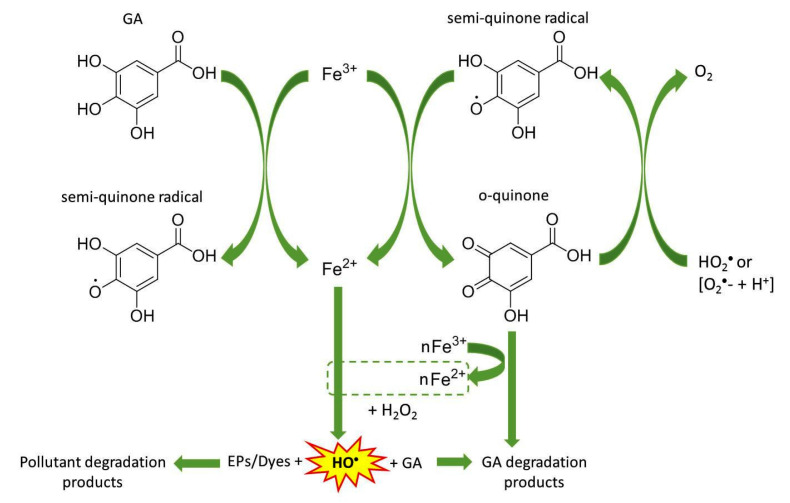
Reduction of Fe^3+^ ions by gallic acid and its oxidized intermediates. The regenerated Fe^2+^ ions can react with H_2_O_2_ to generate more HO^●^ radicals via Fenton reaction. This illustration was adapted from Christoforidis et al. [18], with permission from John Wiley and Sons (ON 5460750316371).

**Table 1 molecules-28-01166-t001:** Advantages and disadvantages of Fenton processes improvised by UV/visible irradiation, ultrasound, and/or electrical current.

Process	Advantages	Disadvantages	Reference
Photo-Fenton	Alternative Fe^2+^ catalyst regeneration. Lower formation of iron sludge. Conducted under natural sunlight (lower cost).	Efficient only for low concentration of organic pollutants. Low utilization of visible light requires UV light for a long time, high energy consumption, and cost.	[22]
Electro-Fenton	Less Fe^2+^ is required, leading to less iron sludge. Avoids purchase, transport, storage, and handling of H_2_O_2_. Process under mild conditions, at room temperature, and atmospheric pressure. Versatility and robustness to automation through electrochemical sensors and devices.	High cost associated with the construction of the electrochemical system. Low activity and stability of the electrodes.	[23]
Sono-Fenton	Increase in mass transfer effects. Regenerates Fe^2+^. Promotes the thermal decomposition of hydrophobic volatile compounds.	High cost and energy-intensive. Need to keep the solution at a low viscosity.	[24]

**Table 2 molecules-28-01166-t002:** Target organic pollutants degradation by Fenton processes in presence of gallic acid.

[Target Pollutant]	Treatment	[Catalyst]	[Oxidant]	[GA]	MolarProportion	pH	Time (Min)	Main Results	Reference
49 μmol.L^−1^ pentachlorophenol	Fe^2+^/H_2_O_2_	9.3 μmol.L^−1^	1550 μmol.L^−1^	235 μmol.L^−1^	1:166.7:25.3	3.5	216	Enhanced degradation in the initial reaction times. Increase in generating HO^●^. When reducing Fe^3+^, GA was degraded in the reactions.	[18]
40 μmol.L^−1^ Methyl orange or Chromotrope 2 R	Fe^2+^/H_2_O_2;_ Fe^3+^/H_2_O_2_	30 μmol.L^−1^	450 μmol.L^−1^	10 μmol.L^−1^	1:15:0.33	2.5–3.0	60	Increase in decolorization and reaction rate constant under different temperatures. Methyl orange decolorization: decrease in activation energy from 81.5 to 53.6 kJ.mol^−1^ with Fe^2+^ and from 102 to 79.1 kJ.mol^−1^ with Fe^3+^. Chromotrope 2 R decolorization: decrease in activation energy from 51.4 to 45.7 kJ.mol^−1^ with Fe^3+^.	[19]
40 μmol.L^−1^ bisphenol A	Fe^3+^/PMS	100 μmol.L^−1^	1 mmol.L^−1^	100 μmol.L^−1^	1:10:1	5.0	10	Increase from ~5% to ~99% in degradation. Increase in reaction rate constant. Increase in the production of HO^●^.	[20]
47.51 mg.kg^−1^ aroclor 1242	Fe^2+^/H_2_O_2_ /Pyrophosphate	10 mmol.L^−1^	100 mmol.L^−1^	10 mmol.L^−1^	1:10:1	7.0	60	Increase from ~50% to ~70% in degradation. Increase in reaction rate constant.	[21]
30 μmol.L^−1^ azure B	Fe^3+^/H_2_O_2_	30 μmol.L^−1^	450 μmol.L^−1^	20 μmol.L^−1^	1:15:0.67	2.7	60	Delay in decolorization. Reduction of Fe^3+^.	[25]
30 μmol.L^−1^ azure B	Cu^2+^/H_2_O_2_	100 μmol.L^−1^	1500 μmol.L^−1^	20 μmol.L^−1^	1:15:0.2	2.7	60	Increase from 20% to 35% in decolorization. Reduction of Cu^2+^.	[25]
20 μmol.L^−1^ iopamidol	Fe^3+^/PS	10 μmol.L^−1^	0.2 mmol.L^−1^	10 μmol.L^−1^	1:20:1	7.0	150	Better mediator among those evaluated. Increase from 10% to 75% in degradation. Increase in reaction rate constant. Increase in generating HO^●^. GA keeps Fe^2+^ available in solution. GA accelerated PS consumption.	[28]
20 μmol.L^−1^ iopamidol	Fe^2+^/PS	10 μmol.L^−1^	0.2 mmol.L^−1^	10 μmol.L^−1^	1:20:1	7.0	150	Increase from ~27% to ~82% in degradation. Increase in reaction rate constant.	[28]
20 μmol.L^−1^ pollutant (Methyl orange, Congo red, or diclofenac)	Fe^3+^/PS	10 μmol.L^−1^	0.2 mmol.L^−1^	10 μmol.L^−1^	1:20:1	7.0	120	Increases from 20% to 58%, 16% to 63%, and 30% to 78% in degrading Methyl orange, Congo red and diclofenac, respectively.	[28]
3.68 μmol.L^−1^ tetrabromobisphenol A	Cu^2+^/ PMS	50 μmol.L^−1^	2 mmol.L^−1^	50 μmol.L^−1^	1:40:1	4.3	10	Increase from 38% to 85% in degradation. Increase in HO^●^ generation. GA was totally degraded. GA reduces Cu^2+^. High concentration of GA inhibited the reactions. Generation of HO^●^, ^1^O_2_, and Cu^3+^.	[29]
2 mg.L^−1^ sulfamethoxazole	Ferrihydrite/ H_2_O_2_	1 g.L^−1^	*In situ* generation	0.2 mmol.L^−1^	-	4.0	120	Increase from 6% to 66% in degradation. Reaction was inhibited at pH 6 and 8. Generation of semiquinone, O_2_^●-^, HO^●^ and ^1^O_2_.	[30]
30 μmol.L^−1^ Bismarck brown Y	Fe^2+^/H_2_O_2_; Fe^3+^/H_2_O_2_	30 μmol.L^−1^	450 μmol.L^−1^	10 μmol.L^−1^	1:15:0.33	2.5–3.0	60	Increase from 78% to 85%, and from 22% to 80% in decolorization by Fe^2+^ and Fe^3+^, respectively. Increase of 15% and 9% in H_2_O_2_ decomposition by Fe^2+^ and Fe^3+^, respectively. Inhibition in decolorization due to increasing concentrations of GA.	[32]
1 mg.L^−1^ 2,2′,5-trichlorodiphenyl (PCB18)	Fe^2+^/H_2_O_2_/Pyrophosphate	5 mmol.L^−1^	100 mmol.L^−1^	100 μmol.L^−1^	1:20:0.02	7.0	480	Low increase in reaction rate constant. Reactions were inhibited by high concentrations of GA.	[33]
50 mg.L^−1^ methyl orange	Fe^3+^/H_2_O_2_	0.5 mmol.L^−1^	10 mmol.L^−1^	0.5 mmol.L^−1^	1:20:1	4.0	5	Increase from 25% to 95% in decolorization. GA was mineralized *in situ*. GA regenerates and keeps Fe^2+^ available in solution through chelation.	[42]
0.412 μmol.L^−1^ 2,2,4,4-tetrabromodiphenylether (BDE47)	Fe^3+^/PMS	13.6 μmol.L^−1^	400 μmol.L^−1^	20.3 μmol.L^−1^	1:29.4:1.49	3.4	4320	Increase in reaction rate constant. Increase from 8% to 85% in degradation. Increase in HO^●^ generation. GA was totally degraded. GA oxidation by-products also reduced Fe^3+^. Reactions were inhibited by high concentrations of GA.	[43]
178 mg.L^−1^ phenol, 364 mg.L^−1^ methomyl, and 30 mg.L^−1^ imidacloprid	Fe^3+^/H_2_O_2_/UV	10 mgl.L^−1^ (0.179 mmol.L^−1^)	200 mgl.L^−1^ (5.88 mmol.L^−1^)	30.6 mg.L^−1^ (0.179 mmol.L^−1^)	1:32.9:1	3.0	120	GA accelerated degradation of each organopollutant in a reaction mixture.	[44]
70 mg.L^−1^ imidacloprid	Fe^3+^/H_2_O_2_/UV	10 mgl.L^−1^ (0.179 mmol.L^−1^)	200 mgl.L^−1^ (5.88 mmol.L^−1^)	30.6 mg.L^−1^ (0.179 mmol.L^−1^)	1:32.9:1	3.0	120	Increase from 22% to 100% in degradation. Fe:GA ratio equal to 1:2 and 1:3 allow a higher amount of soluble iron at pH values above 4.	[44]
0.1 mmol.L^−1^ 2,4-dichlorophenoxyacetic acid	Fe^3+^/H_2_O_2_	1 mmol.L^−1^	10 mmol.L^−1^	1 mmol.L^−1^	1:10:1	6.0	60	GA-Fe complex 1:1 was formed which was then degraded. Complete degradation of the herbicide occurred at 10 min, while the control reaction (with no GA) was negligible. GA was the most effective mediator in decomposing H_2_O_2_.	[45]
0.1 mmol.L^−1^ 2,4,5-trichlorophenoxyacetic acid	Fe^3+^/H_2_O_2_	1 mmol.L^−1^	10 mmol.L^−1^	1 mmol.L^−1^	1:10:1	6.0	1320	Complete degradation of the herbicide occurred at 10 min, while the control reaction was negligible. GA increased the dechlorination. Mineralization of the herbicide increased from ~3% to 65% at 2 h of reaction.	[46]
0.1 mmol.L^−1^ pesticide (atrazine, baygon, carbaryl, or picloram)	Fe^3+^/H_2_O_2_	1 mmol.L^−1^	10 mmol.L^−1^	1 mmol.L^−1^	1:10:1	6.0	120	Complete degradation of the pesticides occurred between 2 and 120 min, while the control reaction removed only from 5% to 10%.	[46]
0.1 mmol.L^−1^ 2,4-dichlorophenoxyacetic acid	Fe^3+^/H_2_O_2_/UV	1 mmol.L^−1^	10 mmol.L^−1^	1 mmol.L^−1^	1:10:1	6.0	120	GA increased the mineralization from ~0% to 73% and 85% in the absence and presence of UV light, respectively.	[46]
100 μmol.L^−1^ 2,4,6-tribromophenol	Fe^3+^-zeolite/H_2_O_2_	109 mg.L^−1^ (30 μmol Fe^3+^.L^−1^)	20 mmol.L^−1^	10 mmol.L^−1^	1:666.7: 333.3	3.0	180	Increase from ~4% to only ~13% in degradation.	[47]
30 μmol.L^−1^ methyl orange	Fe^2+^/H_2_O_2_	30 μmol. L^−1^	450 μmol.L^−1^	10 μmol.L^−1^	1:15:0.33	2.5–3.0	60	Increase from 52% to 95% in decolorization. Increase from 63% to 76% in decomposing H_2_O_2_.	[48]
30 μmol.L^−1^ dye (Methylene blue, Chromotrope 2 R, Methyl orange, or Phenol red)	Fe^3+^/H_2_O_2_	30 μmol.L^−1^	450 μmol.L^−1^	10 μmol.L^−1^	1:15:0.33	2.5–3.0	60	Increase from 27% to 95% in decolorizing Chromotrope 2R. Increase from ~5% to ~40% in decolorizing Phenol red and Methyl orange. Accelerated decolorization of Methylene blue. Increases between 10% and 26% in decomposing H_2_O_2_.	[48]
1 mg.L^−1^ 2,4,4′-trichlorobiphenyl	V^5+^/H_2_O_2_	100 μmol.L^−1^	2 mmol.L^−1^	10 μmol.L^−1^	1:20:0.1	5.5	1440	Better prooxidant among those evaluated. Increase in reaction rate constant. Increase from 43% to 75% in degradation. GA reduces V^5+^ to V^4+^. GA in highest concentration was inhibitory. Increase in degradation and HO^●^ generation at pH 9.0.	[49]
30 μmol.L^−1^ dye (Methylene blue, Chromotrope 2 R, Methyl orange, or Phenol red)	Fe^3+^/H_2_O_2_; Fe^2+^/H_2_O_2_	30 μmol.L^−1^	450 μmol.L^−1^	10 μmol.L^−1^	1:15:0.33	2.5–3.0	60	Increase in reaction rate constants and oxidation capacity.	[50,51]
30 μmol.L^−1^ Bismarck brown Y	Fe^2+^/H_2_O_2;_ Fe^3+^/H_2_O_2_	30 μmol.L^−1^	450 μmol.L^−1^	10 μmol.L^−1^	1:15:0.33	2.5–3.0	60	Increase in decolorization and reaction rate constant under different temperatures. Decrease in activation energy from 75 to 45.9 kJ.mol^−1^ with Fe^2+^ and from 99.5 to 51.1 kJ.mol^−1^ with Fe^3+^.	[52]
48.4 μmol kg^−1^ ibuprofen	Fe^3+^/PS	10 mmol.kg^−1^	100 mmol.kg^−1^	10 mmol.kg^−1^	1:10:1	7.8	120	Increase from ~0% to 81% in degradation.	[53]
6.6 μmol.L^−1^ acetaminophen	Fe^3+^/H_2_O_2_/UV	5 mg.L^−1^ (89.5 μmol.L^−1^)	120 mg.L^−1^ (3.53 mmol.L^−1^)	2.45 mg.L^−1^ (14.4 μmol.L^−1^)	1:39.4:0.16	6.2	5	Increase from ~7% to >95% in degradation. GA increased the solubility of Fe and decomposition of H_2_O_2_.	[54]
6.6 μmol.L^−1^ acetaminophen	Fe^2+^/H_2_O_2_/UV/ electrical current	5 mg.L^−1^(89.5 μmol.L^−1^)	120 mg.L^−1^ (3.53 mmol.L^−1^)	2.45 mg.L^−1^ (14.4 μmol.L^−1^)	1:39.4:0.16	6.2	120	Increase from ~40% to ~100% in degradation. Increase from ~10% to ~55% in soluble Fe content.	[55]

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
