# Peer review of "A Review of Gallic Acid-Mediated Fenton Processes for Degrading Emerging Pollutants and Dyes"

_molecules, 2023, doi:10.3390/molecules28031166_

Round 1

Reviewer 1 Report

Dear Authors,

I have reviewed your Review article “A review about gallic acid-mediated Fenton processes for degrading emerging pollutants and dyes” and expressed my positive feedback regarding your submission. You have explored a very interesting and important topic, but it should be pointed out more, so I am requesting a major revision.

My comments are as follows:

·        In the title, instead of “about”, it should be written “of”

·        In the Abstract, line 12 should introduce the meaning of the abbreviation “GA”.

·        In Table 2 (lines 190, 195, and 197) is some grey box. Check what it should be.

·        The Authors suggest using plant extracts or wood barks containing GA, and other reducers are recommended to make GA-mediated Fenton processes for treating wastewater. But this isn’t elaborated enough. I think the authors should add paragraphs for this part of the research and present it in more detail. The authors mentioned some results in Chapter 3, but poorly, so more information is required.

·        Also, the part about the particular wastewater treatment isn’t elaborated on, it is only mentioned in a few sentences.

Best regards

Author Response

Reviewer 1

I have reviewed your Review article “A review about gallic acid-mediated Fenton processes for degrading emerging pollutants and dyes” and expressed my positive feedback regarding your submission. You have explored a very interesting and important topic;

Reply: We thank the reviewer for the positive and constructive comment.

but it should be pointed out more, so I am requesting a major revision.

My comments are as follows:

- In the title, instead of “about”, it should be written “of”

Reply: Corrected.

- In the Abstract, line 12 should introduce the meaning of the abbreviation “GA”.

Reply: Corrected.

- In Table 2 (lines 190, 195, and 197) is some grey box. Check what it should be.

Reply: Corrected.

- The Authors suggest using plant extracts or wood barks containing GA, and other reducers are recommended to make GA-mediated Fenton processes for treating wastewater. But this isn’t elaborated enough. I think the authors should add paragraphs for this part of the research and present it in more detail. The authors mentioned some results in Chapter 3, but poorly, so more information is required.

Reply: We mentioned in section 3 the main results by using GA-related reductants and plant extracts containing mediators. We believed that the main aspects discussed are enough.

- Also, the part about the particular wastewater treatment isn’t elaborated on, it is only mentioned in a few sentences.

Reply: As the published studies only evaluated synthetic wastewaters (containing only a target pollutant in solution), we highlighted the necessity of evaluating GA-mediated Fenton processes to treat real wastewaters, which contain several pollutants. Anyway, we inserted now other aspects involving real wastewater treatment in the conclusion and perspectives section (lines 379-385).

Reviewer 2 Report

In this work, André Aguiar et al. summarized the recent work on gallic acid-mediated Fenton processes for the degradation of contaminants, this work is interesting and I suggested major revision after addressing these comments:

1.     Line 13, I wonder if it’s better to name it as homogeneous Fenton and heterogeneous Fenton process? The main text should also be revised based on homogeneous and heterogeneous.

2.     Line 14-18, the specific data shouldn’t appear in the abstract of a review. The related data could be summarized in the main text.

3.     These is a question the author should concern, in the complex radical reactions, if (1) the gallic acid could maintain its structure or (2) some of them would be decomposed, or even (3) it would react with the degrading intermediates of the contaminants. I am not sure if the intermediates are toxic, so the authors should search the HPLC and MS spectra of the reported work and write a section on this topic, and finally propose this concern in the prospects.

4.     Line, GA was proved, the English of this work should be well checked.

5.     Line 38, ultrasound and iron salts cannot be the precursors of the radicals, iron salts should be the catalyst for homogeneous Fenton process, of course this is also iron oxides et al. for heterogeneous Fenton (such as J. Bi, Q. Tao, J. Ren, X. Huang, T. Wang, H. Hao, J. Alloys Compd. 937, 2023, 168422. https://doi.org/10.1016/j.jallcom.2022.168422.). Furthermore, the ultrasound was reported could enhanced the generation of radicals, but the most extensively investigated area may be piezocatalysis, and the precursor is not ultrasound, so this sentence should be well arranged.

6.     Line 47, it was always named as Fenton-like reaction.

7.     Line 81-88, the authors should check if there is piezocatalysis.

8.     For a review discussing gallic acid-mediated Fenton processes, I think the contents from Line 1 to Line 93 should be simplified to emphasize the main content of the work.

9.     I wonder if the GA-mediated processes were also integrated with photo-radiation, ultrasound, electric field et al., a small section describing this content is interesting.

10.  For Table 1, strictly, Fenton reaction refers to the iron/H2O2-mediated radical reaction, the authors should think if it’s proper to only use Fenton in the Title and the main text.

11.  Section 2, there are many data and results in the table, except for pH, temperature, dosage et al., the influence of different radical precursors, catalysts, the intensification under UV light, the influence of organic structures, homogeneous or heterogeneous on the removal efficiency should also be discussed, and I suggest the author use a series of sub-titles, such as

(1)   pH

(2)   Temperature

….

(3)   Dosage

(4)   Precursor

(5)   Catalyst form

12.  Line 172-174, references should be provided to support the activation energy.

13.  Line 201, the reason for this phenomenon should be discussed, a review is not a stack of research results.

14.  Line 202-208, I don’t find these results have any relationship with temperature, they are talking about the activation energy after adding GA, it should be discussed in a mechanism section, which is missing in this work.

15.  Line 269, Section 3, it’s a section discussing GA-related molecules not GA, so the title and the related abstract, introduction, discussion, and a table should also be added to summarize the mentioned work.

16.  Section 3, the merits and disadvantages of GA compared with other GA-related molecules should be discussed in a table, and if the authors could discuss their difference from the role of conformation, configuration, functional groups, it would be much better.

17.  Conclusions and future prospects are generally separated contents and there should be at least 2-3 points of prospects.

Author Response

In this work, André Aguiar et al. summarized the recent work on gallic acid-mediated Fenton processes for the degradation of contaminants, this work is interesting and I suggested major revision after addressing these comments:

Reply: We thank the reviewer for the positive and constructive comment.

  1. Line 13, I wonder if it’s better to name it as homogeneous Fenton and heterogeneous Fenton process? The main text should also be revised based on homogeneous and heterogeneous.

Reply: We inserted now this in the abstract, but believed that such an aspect is clarified in the manuscript.

  1. Line 14-18, the specific data shouldn’t appear in the abstract of a review. The related data could be summarized in the main text.

Reply: We preferred to maintain these aspects in the abstract since they represent an important conclusion, regardless of this article is a review.

  1. These is a question the author should concern, in the complex radical reactions, if (1) the gallic acid could maintain its structure or (2) some of them would be decomposed, or even (3) it would react with the degrading intermediates of the contaminants. I am not sure if the intermediates are toxic, so the authors should search the HPLC and MS spectra of the reported work and write a section on this topic, and finally propose this concern in the prospects.

      Reply: This concern is mentioned since the first version of the manuscript, in lines 127-131. As the issue of toxicity was not addressed in the consulted articles, consequently this topic was not detailed in the review. Furthermore, the need for evaluating the toxicity of GA degradation intermediates was mentioned in the conclusions and future perspectives section (lines 374-376).

  1. Line, GA was proved, the English of this work should be well checked.

      Reply: The English have been revised and some sentences were corrected in this new version of the manuscript.

  1. Line 38, ultrasound and iron salts cannot be the precursors of the radicals, iron salts should be the catalyst for homogeneous Fenton process, of course this is also iron oxides et al. for heterogeneous Fenton (such as J. Bi, Q. Tao, J. Ren, X. Huang, T. Wang, H. Hao, J. Alloys Compd. 937, 2023, 168422. https://doi.org/10.1016/j.jallcom.2022.168422). Furthermore, the ultrasound was reported could enhanced the generation of radicals, but the most extensively investigated area may be piezocatalysis, and the precursor is not ultrasound, so this sentence should be well arranged.

Reply: We corrected this sentence to “Different types of AOPs, including photolysis, photocatalysis, ozonation, Fenton reaction, sonochemical and electrochemical oxidation systems, have been recently reviewed for degrading EPs and dyes”.

  1. Line 47, it was always named as Fenton-like reaction.

Reply: Corrected.

  1. Line 81-88, the authors should check if there is piezocatalysis.

Reply: This type of catalysis is uncommon in Fenton-based processes. Few recent papers have approached it.

  1. For a review discussing gallic acid-mediated Fenton processes, I think the contents from Line 1 to Line 93 should be simplified to emphasize the main content of the work.

Reply: We preferred to maintain such aspects detailed in the manuscript.

  1. I wonder if the GA-mediated processes were also integrated with photo-radiation, ultrasound, electric field et al., a small section describing this content is interesting.

Reply: GA was mainly tested with photo-Fenton and the main results are presented in Table 2. We also highlighted this in lines 331-335 since the first version of the manuscript.

  1. For Table 1, strictly, Fenton reaction refers to the iron/H2O2-mediated radical reaction, the authors should think if it’s proper to only use Fenton in the Title and the main text.

Reply: We choose to keep it the way it is, according to other articles of our group.

  1. Section 2, there are many data and results in the table, except for pH, temperature, dosage et al., the influence of different radical precursors, catalysts, the intensification under UV light, the influence of organic structures, homogeneous or heterogeneous on the removal efficiency should also be discussed, and I suggest the author use a series of sub-titles, such as

(1)   pH…

(2)   Temperature….

(3)   Dosage…

(4)   Precursor…

(5)   Catalyst form

Reply: Sub-titles were now created, but their contents are described since the first version of the manuscript.

  1. Line 172-174, references should be provided to support the activation energy.

Reply: These sentences were now excluded in this version of the manuscript since the study referred to did not approach pollutant degradation. Sorry for this.

  1. Line 201, the reason for this phenomenon should be discussed, a review is not a stack of research results.

Reply: We improved these aspects in lines 204-206 in this new version of the manuscript.

  1. Line 202-208, I don’t find these results have any relationship with temperature, they are talking about the activation energy after adding GA, it should be discussed in a mechanism section, which is missing in this work.

      Reply: We improved these aspects in lines 204-206 in this new version of the manuscript.

  1. Line 269, Section 3, it’s a section discussing GA-related molecules not GA, so the title and the related abstract, introduction, discussion, and a table should also be added to summarize the mentioned work.

Reply: A new topic was created about such aspects, which were briefly discussed.

  1. Section 3, the merits and disadvantages of GA compared with other GA-related molecules should be discussed in a table, and if the authors could discuss their difference from the role of conformation, configuration, functional groups, it would be much better.

Reply: We mentioned in section 3 the main results by using GA-related reductants and plant extracts containing mediators. We believed that the main aspects discussed are enough.

  1. Conclusions and future prospects are generally separated contents and there should be at least 2-3 points of prospects.

Reply: This section must not be separated, since it was prepared according to the template. However, improvements were now inserted in this new version onf the manuscript.

Reviewer 3 Report

(1)      “Fenton processes” or “Fenton reaction”, maybe Fenton-like is most suitable.

(2)      The latter four keywords listed should be revised, and “gallic acid” should be mentioned.

(3)      L-cysteine, a green and economical amino acid commonly found in aquatic environment, has been extensively used as reductants in Fenton processes, thus some relevant references, such as 10.1016/j.cej.2022.138588 and 10.1016/j.cclet.2021.10.087 should be cited in introduction.

(4)      The writing style of radicals should be corrected according to the following reference: “Koppenol, W. (2000). "Names for inorganic radicals (IUPAC Recommendations 2000)." Pure and Applied Chemistry 72(3): 437-446.”

(5)      Conclusions and future perspectives should be much more constructive.

Author Response

Dear reviewer,

(1)      “Fenton processes” or “Fenton reaction”, maybe Fenton-like is most suitable.

Reply: Corrected.

(2)      The latter four keywords listed should be revised, and “gallic acid” should be mentioned.

Reply: Gallic acid was included, while the other keywords were maintained.

(3)      L-cysteine, a green and economical amino acid commonly found in aquatic environment, has been extensively used as reductants in Fenton processes, thus some relevant references, such as 10.1016/j.cej.2022.138588 and 10.1016/j.cclet.2021.10.087 should be cited in introduction.

Reply: Cysteine is cited in line 250 since the first version of the manuscript. The second article suggested was inserted also in line 250.

(4)      The writing style of radicals should be corrected according to the following reference: “Koppenol, W. (2000). "Names for inorganic radicals (IUPAC Recommendations 2000)." Pure and Applied Chemistry 72(3): 437-446.” 

Reply: We opted by using the most common names for radicals in Fenton chemistry according to review articles recently published. For e.g. “hydroperoxyl” is more used than “dioxidanidyl”.

(5)      Conclusions and future perspectives should be much more constructive.

Reply: We improved this section in the new version of the manuscript.

Reviewer 4 Report

This manuscript is a short review on the uses of gallic acid for Fenton degradation of organic pollutants in water. Gallic acid has the potential to act as a reducer to improve the Fenton process, given that it is active above the acid pH range required for other commonly used reducers. This manuscript provides a very complete citation of manuscripts dealing with gallic acid, and points to some gaps in the existing research, such as the need for investigating the toxicity of GA systems. This short review is well-written, and it should appeal to a broad audience, and is well-within the scope of this journal. I have no technical comments and recommend the publication without modifications.

Author Response

This manuscript is a short review on the uses of gallic acid for Fenton degradation of organic pollutants in water. Gallic acid has the potential to act as a reducer to improve the Fenton process, given that it is active above the acid pH range required for other commonly used reducers. This manuscript provides a very complete citation of manuscripts dealing with gallic acid, and points to some gaps in the existing research, such as the need for investigating the toxicity of GA systems. This short review is well-written, and it should appeal to a broad audience, and is well-within the scope of this journal. I have no technical comments and recommend the publication without modifications.

Reply: We thank the reviewer for the positive comments.

Reviewer 5 Report

It is an interesting review, it describes the positive effect on the degradation of dyes when using the fenton process with the addition of gallic acid; However, a comparison of efficiencies with the fenton or photo-fenton process is not observed, I mention it because the addition of GA constitutes another factor to control, such as the process of obtaining the extracts and the by-products that would rot. generated at the end of the process, is there no information in this regard that could be incorporated?

Author Response

It is an interesting review, it describes the positive effect on the degradation of dyes when using the Fenton process with the addition of gallic acid;

Reply: We thank the reviewer for the positive comments.

However, a comparison of efficiencies with the fenton or photo-fenton process is not observed, I mention it because the addition of GA constitutes another factor to control, such as the process of obtaining the extracts and the by-products that would rot. generated at the end of the process, is there no information in this regard that could be incorporated?

Reply: We disagree because data presented in Table 2 report the improvements in adding GA when comparing with reaction systems with no mediator. This was highlighted in lines 146-147. The concern with the by-products from GA oxidation is mentioned in lines 127-131 since the first version of the manuscript.

Round 2

Reviewer 1 Report

The authors mostly corrected the manuscript according to the suggestions. Accordingly, I recommend publication in Molecules.

Best regards

Author Response

The authors mostly corrected the manuscript according to the suggestions. Accordingly, I recommend publication in Molecules.

Reply: We thank the reviewer for the positive comment.

Reviewer 2 Report

This work has been improved and reach the standard of the journal, nevertheless, the authors should add the reference electrode information when mention the eletrochemical potential, such as vs. NHE, SHE et al.

Author Response

This work has been improved and reach the standard of the journal, nevertheless, the authors should add the reference electrode information when mention the eletrochemical potential, such as vs. NHE, SHE et al.

Reply: The reviewer is right. We added this information in lines 47 and 50 in this new version of the manuscript.